# Reciprocal regulation among TRPV1 channels and phosphoinositide 3-kinase in response to nerve growth factor

Anastasiia Stratiievska[1], Sara Nelson[1], Eric N Senning[1†], Jonathan D Lautz[2], Stephen EP Smith[2,3], Sharona E Gordon[1]*

[1]Department of Physiology and Biophysics, University of Washington, Seattle, United States; [2]Center for Integrative Brain Research, Seattle Children's Research Institute, Seattle, United States; [3]Department of Pediatrics and Graduate Program in Neuroscience, University of Washington, Seattle, United States

**Abstract** Although it has been known for over a decade that the inflammatory mediator NGF sensitizes pain-receptor neurons through increased trafficking of TRPV1 channels to the plasma membrane, the mechanism by which this occurs remains mysterious. NGF activates phosphoinositide 3-kinase (PI3K), the enzyme that generates $PI(3,4)P_2$ and $PIP_3$, and PI3K activity is required for sensitization. One tantalizing hint came from the finding that the N-terminal region of TRPV1 interacts directly with PI3K. Using two-color total internal reflection fluorescence microscopy, we show that TRPV1 potentiates NGF-induced PI3K activity. A soluble TRPV1 fragment corresponding to the N-terminal Ankyrin repeats domain (ARD) was sufficient to produce this potentiation, indicating that allosteric regulation was involved. Further, other TRPV channels with conserved ARDs also potentiated NGF-induced PI3K activity. Our data demonstrate a novel reciprocal regulation of PI3K signaling by the ARD of TRPV channels.
DOI: https://doi.org/10.7554/eLife.38869.001

*For correspondence:
seg@uw.edu

Present address: †Department of Neuroscience, The University of Texas at Austin, Austin, United States

Competing interests: The authors declare that no competing interests exist.

## Introduction

Although the current opioid epidemic highlights the need for improved pain therapies, in particular for pain in chronic inflammation (*Johannes et al., 2010*), too little is known about the mechanisms that mediate increased sensitivity to pain that occurs in the setting of injury and inflammation (*Ji et al., 2014*). Inflammatory hyperalgesia, the hypersensitivity to thermal, chemical, and mechanical stimuli (*Cesare and McNaughton, 1996*), can be divided in two phases, acute and chronic (*Dickenson and Sullivan, 1987*). Locally released inflammatory mediators, including growth factors, bradykinin, prostaglandins, ATP and tissue acidification (*Kozik et al., 1998*; *Lardner, 2001*; *Tissot et al., 1989*; *Burnstock, 1972*), directly stimulate and sensitize nociceptive fibers of primary sensory neurons (*Cesare and McNaughton, 1996*; *Bevan and Yeats, 1991*; *Trebino et al., 2003*; *Hamilton et al., 1999*; *Mcmahon et al., 1995*).

One of the proteins that has been studied for its role in hyperalgesia is Transient Receptor Potential Vanilloid Subtype 1 (TRPV1). TRPV1 is a non-selective cation channel that is activated by a variety of noxious stimuli including heat, extracellular protons, and chemicals including capsaicin, a spicy compound in chili pepper (*Caterina et al., 1999*). TRPV1 is expressed in sensory nociceptive neurons, which are characterized by cell bodies located in the dorsal root ganglia (DRG) and trigeminal ganglia (*Caterina et al., 1999*). Sensory afferents from these neurons project to skin and internal organs, and synapse onto interneurons in the dorsal horn of the spinal cord (*Willis, 1978*). TRPV1 activation leads to sodium and calcium influx, which results in action potential generation in the sensory neuron and, ultimately, pain sensation (*Caterina et al., 1997*).

The importance of TRPV1 in inflammatory hyperalgesia was demonstrated by findings that the TRPV1 knock-out mouse showed decreased thermal pain responses and impaired inflammation-induced thermal and chemical hyperalgesia (*Caterina et al., 2000*). TRPV1 currents are enhanced during inflammation which leads to increased pain and lowered pain thresholds (*Davis et al., 2000*; *Zhang et al., 2005*; *Shu and Mendell, 1999*). TRPV1 is modulated by G-protein-coupled receptors (GPCRs) and receptor tyrosine kinases (RTKs), but the mechanisms by which these receptors modulate and sensitize TRPV1 are controversial (*Suh and Oh, 2005*; *Shu and Mendell, 1999*; *Cesare and McNaughton, 1996*).

Nerve growth factor (NGF) is one of the best studied RTK agonists involved in inflammatory hyperalgesia (*Vetter et al., 1991*). NGF acts directly on peptidergic C-fiber nociceptors (*Donnerer et al., 1992*), which express RTK receptors for NGF: Tropomyosin-receptor-kinase A (TrkA) (*Mcmahon et al., 1995*) and neurotrophin receptor p75$_{NTR}$ (*Lee et al., 1992*). NGF binding to TrkA/p75$_{NTR}$ induces receptor auto-phosphorylation and activation of downstream signaling pathways including phospholipase C (PLC), mitogen-activated protein kinase (MAPK), and Type IA phosphoinositide 3-kinase (PI3K) (*Vetter et al., 1991*; *Raffioni and Bradshaw, 1992*; *Dikic et al., 1995*). We and others have previously shown that the acute phase of NGF-induced sensitization requires activation of PI3K, which increases trafficking of TRPV1 channels to the PM (*Stein et al., 2006*; *Bonnington and McNaughton, 2003*). In chronic pain, NGF also produces changes in the protein expression of ion channels such as TRPV1 and NaV1.8 (*Ji et al., 2002*; *Thakor et al., 2009*; *Keh et al., 2008*). The acute and chronic phases of the NGF response result in increased 'gain' to painful stimuli.

Type 1A PI3K is a lipid kinase, which phosphorylates the signaling lipids Phosphatidylinositol (4) phosphate (PI4P) and Phosphatidylinositol (4,5) bisphosphate (PIP$_2$) into Phosphatidylinositol (3,4) bis phosphate (PI(3,4)P$_2$)and Phosphatidylinositol (3,4,5) trisphosphate (PIP$_3$), respectively (*Auger et al., 1989*). PI(3,4)P$_2$ and PIP$_3$ are signaling lipids as well, and their role in membrane trafficking and other downstream signaling is well-established (*Insall and Weiner, 2001*; *Hawkins and Stephens, 2016*). PI3K is an obligatory heterodimer that includes the catalytic p110 subunit (with α, β, and γ isoforms) and regulatory p85 subunit (with α and β isoforms) (*Hiles et al., 1992*; *Vanhaesebroeck et al., 2010*). The p85 subunit contains two Src homology 2 (SH2) domains (*Escobedo et al., 1991*), which recognize the phospho-tyrosine motif Y-X-X-M of many activated RTKs and adaptor proteins (*Songyang et al., 1993*). In the resting state, p85 inhibits the enzymatic activity of p110 via one of its SH2 domains (*Miled et al., 2007*). This autoinhibition is relieved when p85 binds to a phospho-tyrosine motif (*Miled et al., 2007*). NGF-induced PI3K activity leads to an increase in the number of TRPV1 channels at the PM (*Bonnington and McNaughton, 2003*; *Stein et al., 2006*).

We have previously shown that TRPV1 and p85 interact directly (*Stein et al., 2006*). We localized the TRPV1/p85 interaction to the N-terminal region of TRPV1 and a region including two SH2 domains of p85 (*Stein et al., 2006*). However, whether the TRPV1/p85 interaction contributes to NGF-induced trafficking of TRPV1 is unknown. Here, we further localized the functional interaction site for p85 to the region of TRPV1 N-terminus containing several conserved Ankyrin repeats (Ankyrin repeat domain (ARD)). Remarkably, we found that TRPV1 potentiated the activity of PI3K and that a soluble TRPV1 fragment corresponding to the ARD was sufficient for this potentiation. Because the ARD is structurally conserved among TRPV channels, we tested whether other TRPV channels could also potentiate NGF-induced PI3K activity. We found that TRPV2 and TRPV4 both potentiated NGF-induced PI3K activity and trafficked to the PM in response to NGF. Together, our data reveal a previously unknown reciprocal regulation among TRPV channels and PI3K. We speculate that this reciprocal regulation could be important wherever TRPV channels are co-expressed with PI3K-coupled RTKs.

## Results and discussion

### NGF-induced trafficking of TRPV1 channels to the PM is preceded by activation of PI3K

To study events that underlie NGF-induced trafficking of TRPV1 to the PM, we used TIRF microscopy to visualize fluorescently labeled TRPV1 (rat) (YFP-fusion, referred to as TRPV1) in transiently

transfected F-11 cells. Cells were also transfected with the NGF receptor subunits TrkA and p75$_{NTR}$ (referred to as TrkA/ p75$_{NTR}$). TIRF microscopy isolates ~ 100 nm of the cell proximal to the coverslip (*Ambrose, 1961*; *Axelrod, 1981*), capturing the PM-proximal fluorescent signals. A change in TRPV1 fluorescence reflects a change in the number of TRPV1 channels at the PM.

We examined NGF-dependent changes in PM-associated TRPV1 before (*Figure 1A*, time point 1), during, and following (*Figure 1A*, time point 2) a 10-min exposure of the cells to NGF (100 ng/ml) via its addition to the bath (bar with gray shading in *Figure 1B,C*). *Figure 1A*, bottom panel shows representative TIRF images of TRPV1 fluorescence of an individual F-11 cell footprint. Consistent with previous findings (*Stein et al., 2006*), upon addition of NGF, TRPV1 levels at the PM increased, with time point two depicting the cell footprint intensity at steady state. For each cell, we normalized the mean fluorescence intensity within the footprint at each time point to the mean between 0 and 60 s prior to the application of NGF. The signal for TRPV1 for the cell in *Figure 1A* is shown in *Figure 1B*, and the collected data, showing the mean and standard error of the mean, are illustrated in *Figure 1C*.

To evaluate NGF-induced PI3K activity, we used fluorescently tagged Btk-PH and Akt-PH, PH (Pleckstrin Homology) domain probes that recognize primarily PIP$_3$ or both PI(3,4)P$_2$ and PIP$_3$, respectively (*James et al., 1996*). Btk-PH and Akt-PH are soluble proteins that localize to the cytosol at rest, when PI(3,4)P$_2$/PIP$_3$ levels are very low, and are recruited to the PM upon stimulation with NGF, when PI3K becomes active and generates PI(3,4)P$_2$/PIP$_3$ (*Hawkins et al., 2006*; *Lemmon, 2008*). A change in PH domain probe fluorescence reflects a change in PI(3,4)P$_2$/PIP$_3$ concentration at the PM, thus serving as an indirect measure of PI3K activity. Because PH domain probes have been reported to interfere with PI3K signaling (*Várnai et al., 2005*), we tested whether Btk-PH and Akt-PH are compatible with NGF signaling to TRPV1. We found that Btk-PH completely abrogated NGF-induced trafficking of TRPV1 to the PM (*Figure 1—figure supplement 1*). In contrast, Akt-PH was fully compatible with NGF-induced trafficking of TRPV1 (*Figure 1*).

As an additional control, we used an orthogonal approach to evaluate the compatibility of the Akt-PH probe with NGF signaling in our cell system. Phosphorylation of the protein kinase Akt (also known as PKB) is a well-studied signaling event downstream of PI3K (*Burgering and Coffer, 1995*; *Kohn et al., 1995*). Akt is phosphorylated in a PI(3,4)P$_2$/PIP$_3$-dependent manner at two sites, T308 and S473, by PDK1 (*Alessi et al., 1997*; *Stokoe et al., 1997*; *Frech et al., 1997*) and mTORC2, respectively (*Sarbassov et al., 2005*). Phosphorylation of Akt at these two sites leads to full activation of Akt, regulating a variety of cellular processes, including the inflammatory response to NGF (*Xu et al., 2007*; *Sun et al., 2007*; *Xu et al., 2011*). Therefore, we tested whether co-expression of the Akt-PH probe altered NGF-induced Akt phosphorylation. We performed western blot analysis using anti-pAKTt308, anti-pAKTs473, and anti-panAKT antibodies. *Figure 1—figure supplement 2* shows that NGF-induced Akt phosphorylation was preserved in cells expressing the Akt-PH probe. We therefore utilized the Akt-PH probe as a readout of PI3K activity in the remaining experiments.

We used two-color TIRF microscopy to measure PI3K activity and TRPV1 trafficking to the PM simultaneously. Treatment of cells with NGF produced an increase in plasma-membrane associated Akt-PH, indicating that PI(3,4)P$_2$/PIP$_3$ levels in the PM increased. The increase was relatively rapid, with kinetics determined by both PI3K activity and the affinity of Akt-PH for PI(3,4)P$_2$/PIP$_3$. The increased Akt-PH signal partially decreased over time even in the continued presence of NGF (*Figure 1B and C* orange, top), possibly due to TrkA/p75$_{NTR}$ receptor internalization (*Grimes et al., 1996*; *Ehlers et al., 1995*) and activation of phosphoinositide 3-phosphatases, e.g. PTEN (*Malek et al., 2017*). NGF treatment also increased the PM TRPV1 signal without an apparent reversal to baseline over the duration of our experiments (*Figure 1B and C* orange, bottom). The peak levels of Akt-PH and TRPV1 for all cells, represented as the normalized intensities measured at 4–6 min (for Akt-PH) and 8–10 min (for TRPV1) after the start of NGF application, are shown in the scatterplot of *Figure 1D*. The distributions were not normal, but skewed toward larger values. This distribution shape is characteristic of NGF-induced TRPV1 sensitization reported previously in DRG neurons (*Stein et al., 2006*; *Bonnington and McNaughton, 2003*), indicating that our cell expression model behaves similarly to isolated DRG neurons. NGF induced a significant increase in Akt-PH levels compared to vehicle (Mean ± SEM: 1.54 ± 0.08, n = 122 compared to 1.01 ± 0.01, n = 32, Wilcoxon rank test p = 10$^{-12}$, *Figure 1C*, top panel, orange and black symbols respectively, see also *Figure 1—figure supplement 3*), and a significant increase in TRPV1 levels compared to vehicle (Mean ± SEM: 1.15 ± 0.02, n = 94 compared to 0.99 ± 0.01, n = 20, Wilcoxon rank test p = 10$^{-6}$;

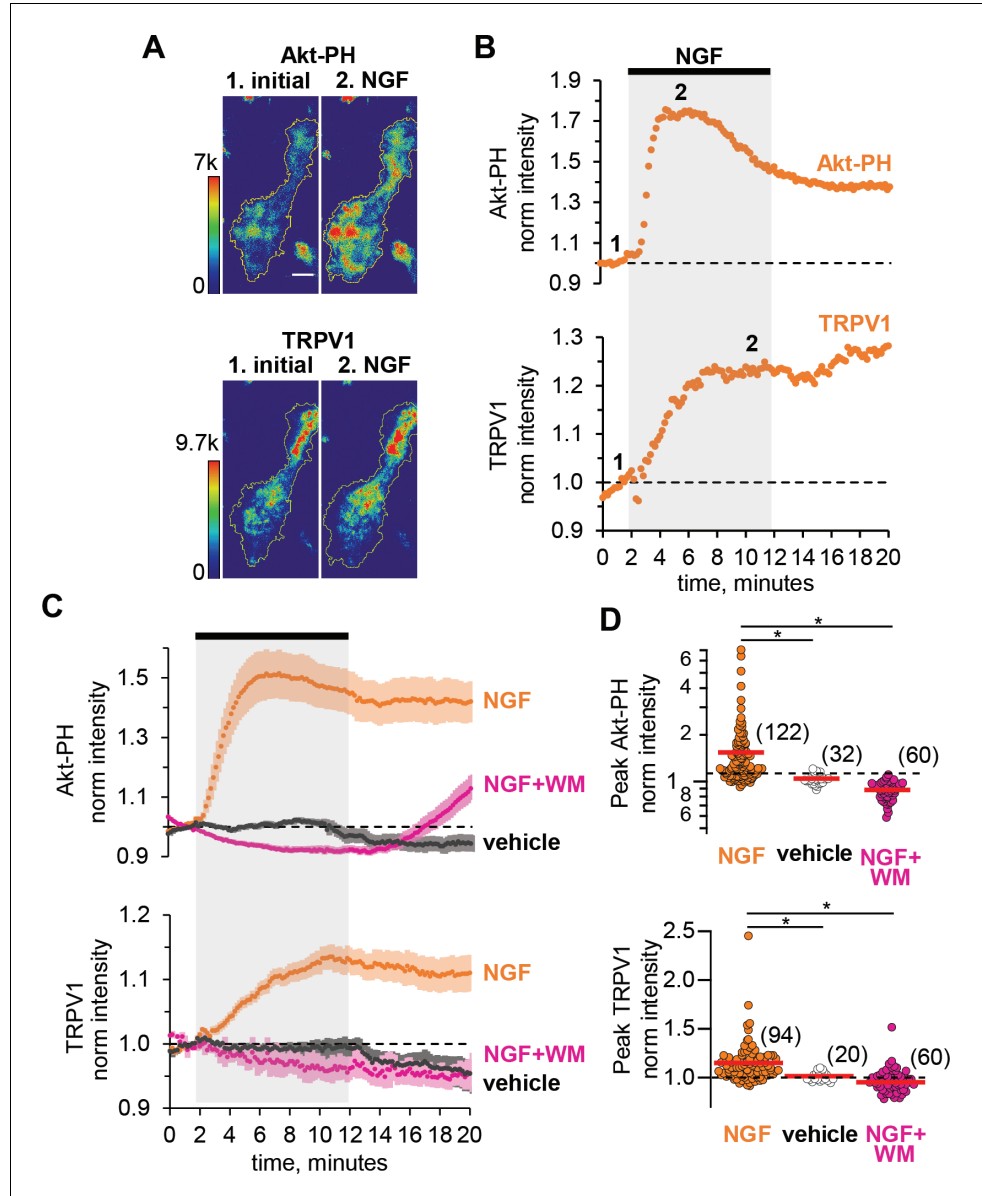

**Figure 1.** NGF increases PIP$_3$ and recruits TRPV1 to the PM. (**A**) TIRF images of a representative F-11 cell transfected with TrkA/p75$_{NTR}$, TRPV1 and Akt-PH. Images labeled one were collected before NGF application and those labeled two were collected at the plateau during NGF application, as indicated by the time points labeled in B. Scale bar is 10 μm. LUT bars represent background-subtracted pixel intensities. The yellow border represents the outline of the cell footprint. (Top) Fluorescence intensity from Akt-PH. (Bottom) Fluorescence intensity from TRPV1. (**B**) Time course of NGF-induced changes in fluorescence intensity for the cell shown in A. NGF (100 ng/mL) was applied during the times indicated by the black bar/gray shading. Intensity at each time point was measured as the mean gray value within the footprint (yellow outline in A). Data were normalized to the mean intensity values during the two minutes prior to NGF application. (**C**) And (**D**) Collected data for the group of cells tested. (**C**) Time course of NGF-induced changes in fluorescence intensity. Averaged time courses of TIRF intensity normalized as in B. Cells treated with either NGF (orange), vehicle (black) or NGF +wortmannin (NGF +WM, magenta), as indicated. TRPV1 (bottom) and Akt-PH (top). Error bars are SEM (**D**) NGF-induced change in fluorescence intensity. Cells were treated with NGF (orange), vehicle (open symbols) or NGF +wortmannin (NGF +WM, magenta), as indicated. Averaged normalized TIRF intensity during NGF application (6–8 min for Akt-PH (top) and 10–12 min for TRPV1 (bottom)). The red bars indicate the mean Akt-PH fluorescence (top) and TRPV1 fluorescence (bottom). Asterisks indicate Wilcoxon rank test significance p value < 0.001.
DOI: https://doi.org/10.7554/eLife.38869.002

*Figure 1 continued on next page*

*Figure 1 continued*

The following source data and figure supplements are available for figure 1:

**Figure supplement 1.** Btk-PH is not compatible with NGF signaling to TRPV1.

DOI: https://doi.org/10.7554/eLife.38869.003

**Figure supplement 2.** Akt-PH expression does not interfere with NGF-induced Akt phosphorylation.

DOI: https://doi.org/10.7554/eLife.38869.004

**Figure supplement 2—source data 1.** Full images of gel in *Figure 1—figure supplement 2*.

DOI: https://doi.org/10.7554/eLife.38869.007

**Figure supplement 3.** Vehicle does not increase PIP$_3$ or recruit TRPV1 to PM.

DOI: https://doi.org/10.7554/eLife.38869.005

**Figure supplement 4.** Model for TIRF illumination and estimation of Akt-PH translocation to the PM.

DOI: https://doi.org/10.7554/eLife.38869.006

**Figure supplement 4—source data 1.** Depth of TIRF field and membrane translocation estimation.

DOI: https://doi.org/10.7554/eLife.38869.020

---

*Figure 1C*, bottom panel, orange and black symbols respectively, see also *Figure 1—figure supplement 3*). Consistent with a PI3K-dependent mechanism, the NGF-induced increases in both PM-associated Akt-PH and TRPV1 were prevented by the PI3K inhibitor wortmannin (20 nM) (*Figure 1C and D*, magenta, n = 60, Mean ±SEM for Akt-PH – 0.88 ± 0.01 and for TRPV1 – 0.95 ± 0.01; Wilcoxon rank test p value for Akt-PH – $10^{-13}$ and for TRPV1 – $10^{-10}$).

TIRF microscopy is often discussed as a method that isolates a fluorescence signal at the PM (*Axelrod, 1981*). Indeed, illumination falls off exponentially with distance from the coverslip (*Ambrose, 1961*). Nevertheless, with a typical TIRF setup such as that used for this study (see Materials and methods) ~90% of the signal comes from the cytosol (*Figure 1—figure supplement 4*, also see Materials and methods), assuming the incident light was at the critical angle and that the membrane bilayer and associated protein layer extends up to ~10 nm from the coverslip. The contamination of the TIRF signal with fluorescence from the cytosol leads to an underestimation of the change in PM-associated fluorescence from Akt-PH and TRPV1. Under our experimental conditions, we estimate that the ratio of the total fluorescence intensity measured after and before NGF application, $F_{NGF}$, of 1.54 translates into about a 10-fold increase in PM-associated fluorescence, $R_m$ (*Figure 1—figure supplement 4*; see Materials and methods), although this should be considered just a rough estimate.

## TRPV1 potentiates NGF-induced PI3K activity

Comparing the NGF-induced increase in Akt-PH in control cells that did not express TRPV1 to that in cells expressing TRPV1, we made an unexpected observation: TRPV1 appeared to potentiate NGF-induced PI3K activity. Comparing the time course of the NGF response in cells without TRPV1 (*Figure 2A*, blue trace) to cells expressing TRPV1 (*Figure 2A*, orange), we found a pronounced increase in Akt-PH fluorescence intensity in TRPV1-expressing cells. This increase was statistically significant, with the peak normalized Akt-PH intensity value of 1.08 ± 0.03 (n = 75) in cells without TRPV1 and 1.54 ± 0.08 (n = 122) in cells expressing TRPV1 (*Figure 2B*, Wilcoxon rank test p = $10^{-12}$, see also *Figure 2—figure supplement 1A*). Interestingly, the dynamics of NGF-induced PI(3,4)P$_2$/PIP$_3$-generation in the absence of TRPV1 were also different in that PI(3,4)P$_2$/PIP$_3$ levels were sustained. As in TRPV1-expressing cells, the NGF-induced increases in PI(3,4)P$_2$/PIP$_3$ levels in control cells were prevented by treatment of cells with wortmannin (*Figure 2—figure supplement 2*, Mean ± SEM: 0.81 ± 0.02, n = 53; Student's t-test p-value was $10^{-16}$).

One possible cause for the potentiation of NGF-induced PI3K activity we observed in TRPV1-expressing cells could be a change in PI3K expression levels in TRPV1 vs. control cells. To determine whether this was the case, we performed western blot analysis with an anti-p85α antibody to quantify the PI3K protein levels across transfection conditions. As shown in *Figure 2—figure supplement 3A*, expression of TRPV1 did not alter the expression level of the p85α subunit of PI3K. We quantified protein expression levels using densitometry, and normalized expression to tubulin, giving the relative expression levels shown in *Figure 2—figure supplement 3B*. Average relative p85α expression levels were similar between non-TRPV1 expressing cells and cells expressing TRPV1 (n = 5, Student's t-test p value was 0.95). We conclude that a difference in PI3K expression in TRPV1-

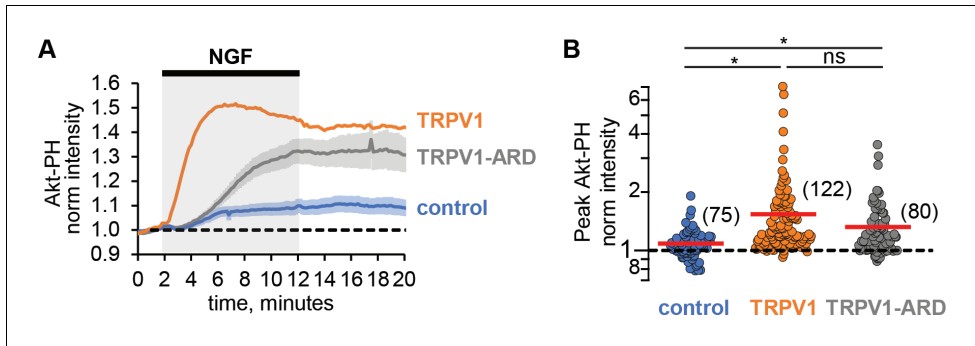

**Figure 2.** TRPV1-ARD is necessary and sufficient for potentiation of NGF-induced PI3K activity. (**A**) Time course of NGF-induced changes in Akt-PH fluorescence intensity. NGF (100 ng/mL) was applied during the times indicated by the black bar/gray shading. Averaged normalized TIRF intensity from cells transfected with TrkA/p75$_{NTR}$ and Akt-PH: control cells without TRPV1 (blue, n = 75), TRPV1 (orange, n = 122), or TRPV1-ARD (gray, n = 80). Traces represent the mean and error bars represent the SEM. TRPV1 data are the same as in *Figure 1C*, error bars removed for clarity. (**B**) NGF-induced changes in Akt-PH fluorescence intensity for control cells (blue), cells expressing TRPV1 (orange data are the same as in *Figure 1D*) and cells transfected with TRPV1-ARD (gray). Averaged normalized TIRF intensity during NGF application (6–8 min). Red bars indicate mean (see *Table 2* for values). Asterisks indicate significance of Holm-Bonferroni post-hoc adjusted Wilcoxon rank test p value < 0.001 (see *Table 2* for values).

DOI: https://doi.org/10.7554/eLife.38869.008

The following source data and figure supplements are available for figure 2:

**Figure supplement 1.** Representative images of NGF-induced recruitment Akt-PH and TRP channels to the PM.
DOI: https://doi.org/10.7554/eLife.38869.009

**Figure supplement 2.** PI(3,4)P$_2$/PIP$_3$ generation is diminished by PI3K inhibitor wortmannin.
DOI: https://doi.org/10.7554/eLife.38869.010

**Figure supplement 3.** TRPV1 co-expression does not alter PI3K expression.
DOI: https://doi.org/10.7554/eLife.38869.011

**Figure supplement 3—source data 1.** Full image of gel in *Figure 2—figure supplement 3*.
DOI: https://doi.org/10.7554/eLife.38869.012

expressing vs. control cells did not account for the observed TRPV1-induced potentiation of NGF-stimulated PI3K activity.

## The ARD of TRPV1 is sufficient for potentiation of NGF-induced PI3K activity

We have previously shown that the N-terminal region of TRPV1, consisting of 110 amino acids and the ankyrin repeat domain (TRPV1-ARD), interacts directly with the p85 subunit of PI3K in yeast two-hybrid assays, co-immunoprecipitation from cells, and using recombinant fragments in vitro (*Stein et al., 2006*). We hypothesized that the TRPV1-ARD might also mediate NGF-induced potentiation of PI3K. To determine whether the ARD is sufficient for potentiation of NGF-induced PI3K activity, we expressed the ARD as a fragment and then measured NGF-induced PI3K activity. As shown in *Figure 2A* (gray trace), NGF induced PI3K activity that was greater in TRPV1-ARD expressing cells than in control cells (blue trace). The increase in peak Akt-PH normalized intensity was statistically significant compared to control cells, with a mean of 1.32 (±0.02, n = 80; *Figure 2B*; Wilcoxon rank test p = $10^{-5}$, see also *Figure 2—figure supplement 1B*). The kinetics of this potentiation were somewhat slower with TRPV1-ARD compared to TRPV1 (*Figure 2A*, orange trace), so that Akt-PH reached steady-state levels somewhat later during NGF treatment. Nevertheless, the potentiation of NGF-induced PI3K activity by the ARD fragment was nearly as great as observed with full-length TRPV1 (Wilcoxon rank test p = 0.08). In addition, the ability of a soluble TRPV1 fragment to reconstitute potentiation suggests that the mechanism of potentiation is at least partly allosteric, involving more than just a tethering of PI3K at the membrane by TRPV1.

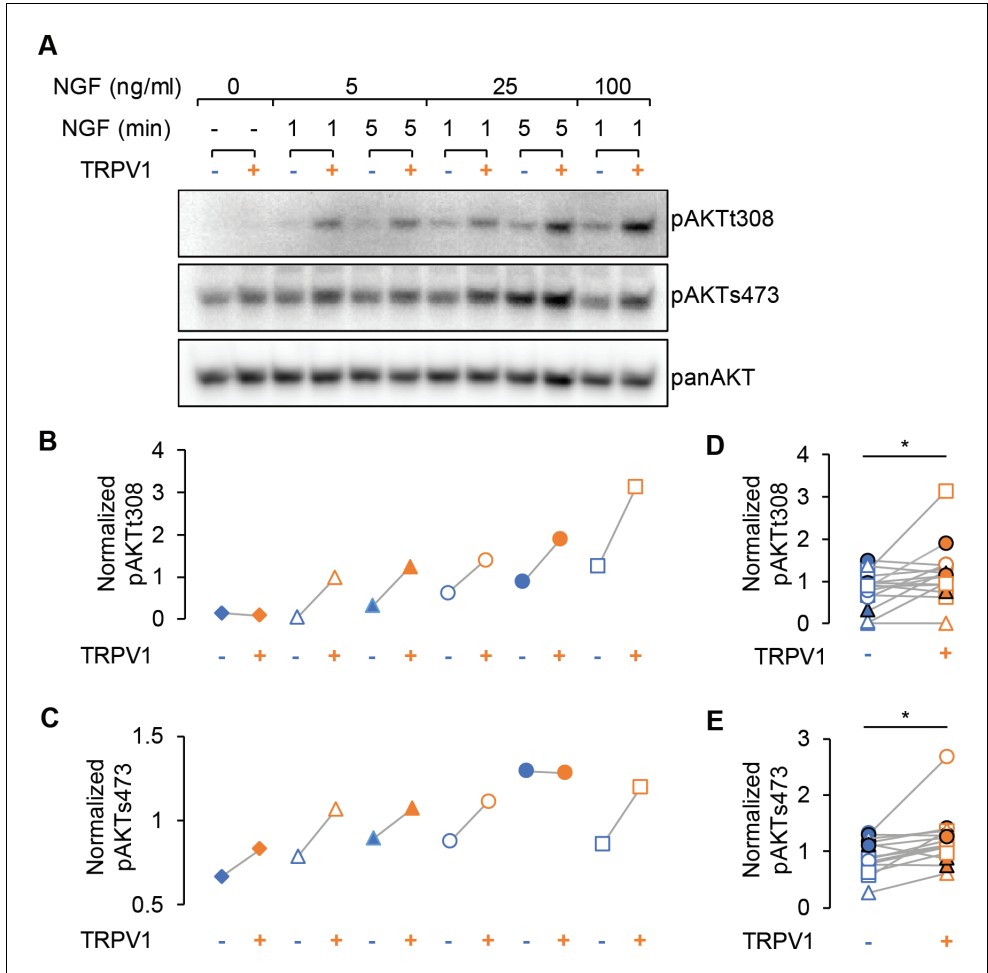

**Figure 3.** TRPV1 enhances NGF-induced Akt phosphorylation. (**A**) Representative immunoblot staining for analysis of Akt phosphorylation in F-11 cells transfected same as in imaging experiments. Cells were treated with indicated dose of NGF for an indicated amounts of time, lysed and loaded on SDS-PAGE. The same membrane was probed with pAKTs473, stripped and re-probed with pAKTt308 and again with panAKT antibodies (see Materials and methods). (**B**) and (**C**) Analysis of the representative blots shown in (**A**). Each band average intensity was normalized to the average of the blot and then divided by that of the corresponding lane of the panAkt blot. Akt phosphorylated at T308 (**B**) and S473 (**C**) from control cells (blue symbols) and cells expressing TRPV1 (orange symbols) treated with NGF (5, 25 or 100 ng/ml) for 1 or 5 min as indicated in (**A**). Triangles represent treatment with NGF 5 ng/ml, circles – 25 ng/m, squares – 100 ng/ml. Open symbols represent treatments for 1 min and filled symbols – 5 min. (**D**) and (**E**) Normalized phospho-Akt intensities from all indicated conditions are pooled together for the n = 3 of independent experiments. Paired Student's t-test for pAKTt308 p=0.02 and for pAKTs473 p=0.008.
DOI: https://doi.org/10.7554/eLife.38869.013

The following source data is available for figure 3:

**Source data 1.** Full images of gels in *Figure 3*.
DOI: https://doi.org/10.7554/eLife.38869.016

## An orthogonal PI3K assay confirms that TRPV1 potentiates NGF-induced PI3K activity and generation of PI(3,4)P$_2$/PIP$_3$

We used the Akt phosphorylation assay described above as an orthogonal method of examining the potentiation of NGF-induced PI3K activity in TRPV1-expressing cells. We performed western blot analysis using phospho-specific Akt antibodies, reprobing the blots with a pan Akt antibody for normalization purposes (*Figure 3A*). Because phosphorylation at T308 and S473 are differentially regulated, we used three concentrations of NGF (5, 25, and 100 ng/mL) and two incubation times (1 and 5 min). We observed increased phosphorylation at both T308 and S473 in TRPV1-expressing cells

compared to control cells for almost all trials with all three NGF concentrations and both time points (*Figure 3B,C*). The enhanced NGF-induced Akt phosphorylation was statistically significant for both T308 and S473 sites for all conditions pooled together (*Figure 3D,E*; paired Student's t-test for T308 p = 0.02 and S473 p = 0.008). Thus, TRPV1 potentiation of NGF-induced PI3K activity is sufficient to enhance $PI(3,4)P_2$ and/or $PIP_3$ levels to increase Akt phosphorylation.

Finally, *Figure 3* shows that the extent of Akt phosphorylation in unstimulated cells was indistinguishable in control vs. TRPV1-expressing cells at both S308 ($Intensity_{pAkt/pan\ Akt}$: 0.075 ± 0.004 for control and 0.076 ± 0.004 for TRPV1, Mean ± SEM, n = 3, paired Student's t-test p = 0.95) and T473 sites ($Intensity_{pAkt/pan\ Akt}$: 0.3 ± 0.24 for control and 0.23 ± 0.14 for TRPV1, Mean ± SEM, n = 3, paired Student's t-test p = 0.44), indicating that TRPV1 did not perturb the levels of $PI(3,4)P_2/PIP_3$ at rest. Importantly, we examined whether NGF-induced phosphorylation at both T308 and S473 required expression of TrkA/p75_{NTR}. NGF-induced phosphorylation of Akt was not observed in cells in which TrkA/p75_{NTR} were not expressed (*Figure 1—figure supplement 2*). Together with the data using Akt-PH in TIRF microscopy experiments, these data indicate that NGF-induced PI3K activity is greater, and $PI(3,4)P_2/PIP_3$ production is greater, in TRPV1-expressing cells than in those that do not express TRPV1.

## Potentiation of PI3K and NGF-induced trafficking are conserved among TRPV channels

The ARD of TRPV1 is highly conserved among other members of the TRPV family of ion channels (*Gaudet, 2008*). Given the sufficiency of the TRPV1 ARD in potentiation of NGF-induced PI3K activity, we postulated that reciprocal regulation among other TRPV family members and PI3K would occur as well. We examined whether other ARD-containing TRP channels, TRPV2 (rat) and TRPV4 (human) were trafficked to the plasma membrane in response to NGF. Using TRPV2 and TRPV4 fused to fluorescent proteins, we found that they were both trafficked to the PM in response to NGF compared to vehicle (Holm-Bonferroni post-hoc adjusted Wilcoxon rank test p < 0.05 see *Table 1*, *Figure 4C,D*, see also *Figure 4—figure supplement 1* for representative images). In addition, we found that the NGF-induced increase in Akt-PH was significantly greater in TRPV2- and TRPV4-expressing cells compared to control cells (Holm-Bonferroni post-hoc adjusted Wilcoxon rank test p < 0.05 see *Table 2*, *Figure 4A,B*). The effects of TRPV2 and TRPV4 on $PI(3,4)P_2/PIP_3$ levels were significantly smaller than those elicited by TRPV1 (Holm-Bonferroni post-hoc adjusted Wilcoxon rank test p < 0.05 see *Table 2*). Further experiments would be required to determine whether the differences were due to differences in expression level, differences in the affinity of PI3K for the TRPV ARDs, or differences in the effect of each ARD on the catalytic activity of PI3K. We conclude that potentiation of NGF-induced PI3K activity and traffic to the PM in response to NGF are conserved among TRPV1, TRPV2, and TRPV4.

Increased trafficking of TRPV1 to the cell surface is essential for sensitization to noxious stimuli produced by NGF and other inflammatory mediators (*Morenilla-Palao et al., 2004*; *Ferrandiz-Huertas et al., 2014*). Although the involvement of PI3K in NGF-induced sensitization has been known for over a decade (*Bonnington and McNaughton, 2003*; *Stein et al., 2006*), the role, if any,

**Table 1.** Normalized TRP channel fluorescence intensities measured during NGF application for all discussed conditions.

The number of cells in the data set collected over at least three different experiments is given by n. Non-adjusted Wilcoxon rank test two tail p values was performed for pairwise comparisons as indicated.

| | NGF Mean ± SEM | N= | TRPV1 | Vehicle |
|---|---|---|---|---|
| TRPV1 | 1.15 ± 0.02 | 94 | - | - |
| vehicle | 1.01 ± 0.01 | 20 | $10^{-6}$ | - |
| TRPV2 | 1.12 ± 0.02 | 62 | 0.24 | 0.002 |
| TRPV4 | 1.11 ± 0.02 | 48 | 0.13 | 0.002 |

DOI: https://doi.org/10.7554/eLife.38869.017

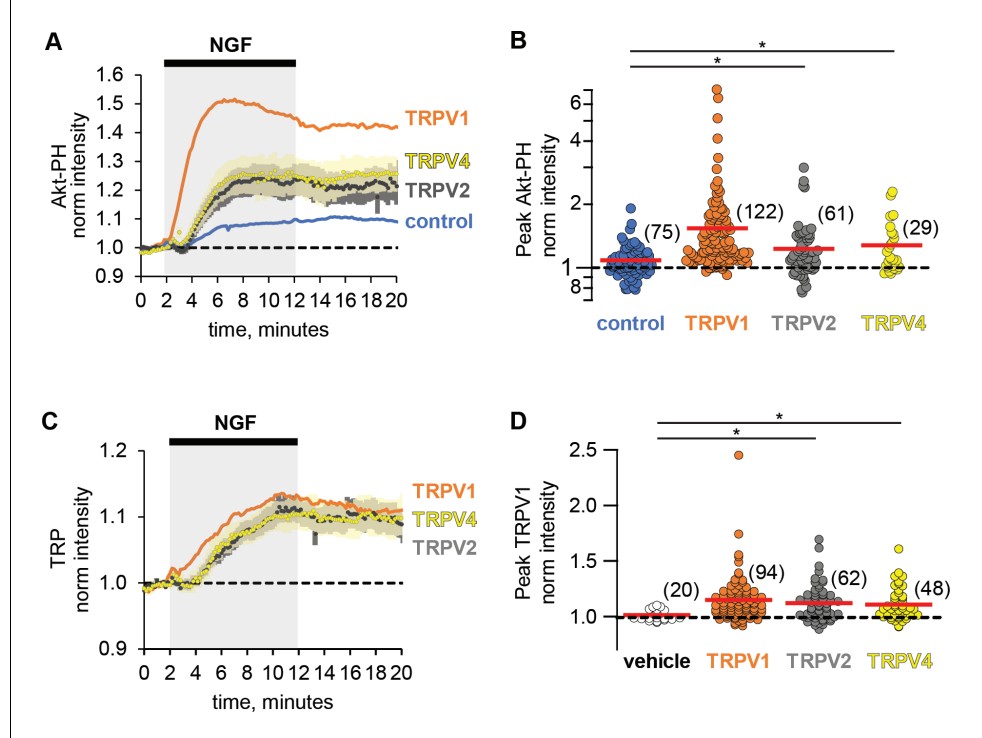

**Figure 4.** Potentiation of PI3K and NGF-induced trafficking are conserved among TRPV channels. Time course of NGF-induced changes in fluorescence intensity. NGF (100 ng/mL) was applied during the times indicated by the black bar/gray shading. Traces represent the mean, error bars are SEM. Control and TRPV1 data same as in *Figure 2* with error bars removed for clarity. (**A**) Averaged normalized TIRF intensity of Akt-PH from cells transfected with TrkA/p75$_{NTR}$ and Akt-PH and: (**A**) no channel (control; blue; n = 75); TRPV1 (orange; n = 122); TRPV2 (black; n = 61); TRPV4 (yellow; n = 29). (**B**) Averaged normalized Akt-PH intensity during NGF application (6–8 min). The red bars indicate the mean. Asterisks indicate significance (Holm-Bonferroni post-hoc adjusted Wilcoxon rank test p < 0.05, see *Table 2* for values). (**C**) Averaged normalized TIRF intensity of individual TRP channels. Color scheme as in (**A**) with the cell numbers as follows: TRPV1 (n = 94); TRPV2 (n = 62); TRPV4 (n = 48). (**D**). Averaged normalized TRP channel intensity during NGF application (8–10 min). The red bars indicate the mean. Asterisks indicate significance (Holm-Bonferroni post-hoc adjusted Wilcoxon rank test p < 0.05, see *Table 1* for values).

DOI: https://doi.org/10.7554/eLife.38869.014

The following figure supplement is available for figure 4:

**Figure supplement 1.** Representative images of NGF-induced recruitment Akt-PH and TRP channels to the PM.
DOI: https://doi.org/10.7554/eLife.38869.015

of direct binding of TRPV1 and PI3K was unclear. Here, we show that ARD region of TRPV1 that binds PI3K is sufficient to potentiate NGF-induced PI3K activity. Although it is possible that TRPV1 inhibition of the PI(3,4)P$_2$/PIP$_3$ phosphatase PTEN (*Malek et al., 2017*) could contribute to TRPV1 potentiation of NGF-induced increases in PI(3,4)P$_2$/PIP$_3$ levels, this and other more complex models are not needed to explain our data. In addition, whereas the present work does not rule out that the potentiation of PI3K we describe requires an effector that mediates signaling between the TRPV1 ARD and PI3K, we favor a simpler model in which the previously described direct interaction between TRPV1 and PI3K mediates the signaling. We speculate that, without TRPV1 potentiation of PI3K, NGF signaling would not produce sufficient PI(3,4)P$_2$/PIP$_3$ to promote channel trafficking during inflammation. Future studies that decouple potentiation of PI3K activity from the expression of TRPV channels will be needed to determine whether the reciprocal regulation between ARD-containing TRPV channels and PI3K serves an obligate role in channel sensitization.

Is reciprocal regulation among TRPV channels and PI3K relevant beyond pain signaling? TRPV channels have been proposed to be involved in RTK/PI3K signaling in a variety of cell types

**Table 2.** Normalized Akt-PH fluorescence intensities measured during NGF application for all discussed conditions.

The number of cells in the data set collected over at least three different experiments is given by n. Non-adjusted Wilcoxon rank test two tail p values for pairwise comparisons as indicated.

| Akt-PH from | NGF Mean ± SEM | N= | Control | TRPV1 |
|---|---|---|---|---|
| control | 1.08 ± 0.03 | 75 | - | - |
| TRPV1 | 1.54 ± 0.8 | 122 | $10^{-12}$ | - |
| TRPV1-ARD | 1.32 ± 0.2 | 80 | $10^{-5}$ | 0.08 |
| TRPV2 | 1.23 ± 0.18 | 61 | 0.04 | 0.0002 |
| TRPV4 | 1.28 ± 0.14 | 29 | 0.02 | 0.02 |

DOI: https://doi.org/10.7554/eLife.38869.018

(*Reichhart et al., 2015*; *Katanosaka et al., 2014*; *Jie et al., 2015*; *Sharma et al., 2017*). For example, TRPV2 is co-expressed in muscle cells with the insulin like growth factor receptor (IGFR) and is known to be important in muscle loss during muscular dystrophy (*Iwata et al., 2003*). The mechanism is believed to involve IGFR activation leading to increased trafficking of TRPV2 to the sarcolemma, $Ca^{2+}$ overload/cytotoxicity, and cell death (*Iwata et al., 2003*; *Perálvarez-Marín et al., 2013*; *Katanosaka et al., 2014*). Whether TRPV2 potentiates IGF-induced PI3K activity remains to be determined. The co-expression of TRPV channels with RTK/PI3K in other tissues, including nerve (TRPV1/NGF) (*Tanaka et al., 2016*), muscle (TRPV2/IGF) (*Katanosaka et al., 2014*) and lung (TRPV4/TGFβ1) (*Rahaman et al., 2014*) raises the question of whether reciprocal regulation among TRPV channels and PI3K plays a role in RTK signaling in cell development, motility, and/or pathology.

## Materials and methods

### TIRF microscopy and analysis

For imaging, we used an inverted microscope (NIKON Ti-E) equipped for total internal fluorescence (TIRF) imaging with a 60x objective (NA 1.49). Glass coverslips with adherent cells were placed in a custom-made chamber. The chamber volume (~1 ml) was exchanged using a gravity-driven perfusion system. Cells were acclimated to flow for at least 15 min prior to NGF application. Akt-PH fused to Cyan Fluorescent Protein (CFP) was imaged using excitation from a 447 nm laser and a 480/40 emission filter. TRPV1 fused to Yellow Fluorescent Protein (YFP) was imaged using the 514 nm line of an argon laser and a 530 long-pass emission filter. Time-lapse images were obtained by taking consecutive CFP and YFP images every 10 s. Movies were then processed using ImageJ software (NIH) (*Rasband, 1997*). Regions of interest (ROI) were drawn around the footprint of individual cells and the average ROI pixel intensity was measured. Measurements were analyzed using Excel 2013 (Microsoft Corporation), by subtracting the background ROI intensity from the intensity of each cell ROI. Traces were normalized by the average intensity during the 1-min time period prior to NGF application.

### Depth of TIRF field and membrane translocation estimation

Because $PI(3,4)P_2/PIP_3$ levels reported by the Akt-PH fluorescence measured with TIRF microscopy include significant contamination from free Akt-PH in the cytosol, we used the characteristic decay of TIRF illumination to estimate the fraction of our signal due to Akt-PH bound to the membrane. We first estimated the fraction of the illumination at the membrane in resting cells, assuming that free Akt-PH is homogeneously distributed throughout the evanescent field. After stimulation with NGF, we then used this fraction of illumination at the membrane to determine the fraction of the emission light originating from this region. The estimation approach used below was not used to quantitatively evaluate our data. Rather, it demonstrates the general issue of cytosolic contamination causing underestimation of changes in membrane-associated fluorescence even when using TIRF microscopy.

The depth of the TIRF field was estimated as described in the literature (*Axelrod, 1981*; *Mattheyses and Axelrod, 2006*). Briefly, when laser light goes through the interface between a

coverslip with refractive index $n_2$ and saline solution with refractive index $n_1$, it experiences total internal reflection at angles less than the critical incidence angle, $\theta_c$, given by

$$\theta_c = sin^{-1}\left(\frac{n_1}{n_3}\right)$$

The characteristic depth of the illuminated field $d$ is described by

$$d = \frac{\lambda_0}{4\pi n_3}\left(sin^2\theta - sin^2\theta_c\right)^{-\frac{1}{2}}$$

where $\lambda_0$ is laser wavelength. The illumination decay $\tau$, depends on depth of field as follows:

$$\tau = \frac{1}{d}$$

TIRF illumination intensity, $I$, is described in terms of distance from the coverslip, $h$, by

$$I = e^{-\tau h}$$

For simplicity, we measured the distance $h$ in 'layers', with the depth of each layer corresponding to physical size of Akt-PH, which was estimated to be approximately 10 nm based on the sum of longest dimensions of Akt-PH and GFP in their respective crystal structures (PDB ID: 1UNQ and 1GFL). We solved for TIRF illumination intensity using the following values for our system: refractive indexes of solution $n_1$ = 1.33 and coverslip $n_3$ = 1.53, critical incidence angle $\theta_C$ = 60.8 degrees. The laser wavelength used in our experiments was $\lambda_0$ = 447 nm, and the experimental angle of incidence was $\theta_{exp}$ = 63 degrees. This produces a characteristic depth of $d_{63}$ = 127 nm and an illumination decay of $\tau_{63}$ = 0.008 nm$^{-1}$. We plot TIRF illumination intensity over distance in molecular layers and nanometers in *Figure 1—figure supplement 4*.

The values determined above allow us to estimate the contributions to our TIRF signal from the membrane vs. the cytosol. According to our calculation, the TIRF illumination intensity approaches 0 at around 500 nm, or layer $h_{49}$. We consider the membrane and associated proteins to reside in layer $h_0$. Under these conditions, at rest, 5% of total recorded TIRF fluorescence arises from $h_0$, with the remainder originating from $h_1$-$h_{49}$. At rest, we assume that Akt-PH molecules are distributed evenly throughout layers $h_0$-$h_{49}$, with no Akt-PH bound to the membrane because the concentration of PI (3,4)P$_2$/PIP$_3$ in the PM is negligible at rest. Total fluorescence intensity measured before NGF application, $F_{initial}$, depends on $m$, the number of molecules per layer at rest, B, the brightness of a single molecule of CFP, and TIRF illumination intensity, $I$:

$$F_{initial} = B * \sum_{0}^{49} mI_i$$

Normalizing our time traces to $F_{initial}$, sets $F_{initial}$ = 1. We solved for $m$ numerically using Excel (Microsoft, Redmond, WA; see *Figure 1—figure supplement 4—source data 1*), and determined a value of 0.08. We assumed a fixed number of molecules in the field and that the only NGF-induced change was a redistribution of molecules among layers. The total fluorescence intensity measured after NGF application, $F_{NGF}$, will reflect the redistribution of $\Delta m$ molecules between membrane layer $h_0$ and all layers $h_0$-$h_{49}$, with free Akt-PH homogeneously distributed among these layers. Therefore, $F_{NGF}$ is a sum of fluorescence intensities of the number of bound molecules in the membrane layer $h_0$ and the free molecules in layers $h_1$-$h_{49}$:

$$F_{NGF} = B * \left[(\Delta m)I_0 + \sum_{0}^{49}\left(m - \frac{\Delta m}{50}\right)I_i\right]$$

We solved for $\Delta m$ using Excel, constraining $F_{NGF}$ to the values we measured for control and TRPV1-expressing cells (data listed in the table in *Figure 1—figure supplement 4B*). Finally, we estimated the NGF-induced change in Akt-PH bound to the membrane as $R_m$, the ratio of molecules in $h_0$ after NGF to that before NGF:

$$R_m = \frac{(m + \Delta m)I_0}{mI_0}$$

We compared $R_m$ values to the $F_{NGF}$ values listed in the table *Figure 1—figure supplement 4B*. For example, in cells expressing TRPV1, $F_{NGF}$ of 1.54 led to 10 times more membrane-associated Akt-PH molecules. Note, that if we instead allow the number of molecules in cytosolic layers to remain constant as $m_0$ increases with NGF treatment, we calculate an $R_m$ value of 8, very similar to the value of 10 obtained with redistribution of a fixed number of molecules across all layers. Both of these scenarios are independent of the initial Akt-PH fluorescence intensity in a given cell.

## Cell culture/transfection/ DNA constructs/solutions

F-11 cells (a gift from M.C. Fishman, Massachusetts General Hospital, Boston, MA; (*Francel et al., 1987*)) were cultured at 37°C, 5% $CO_2$ in Ham's F-12 Nutrient Mixture (#11765–054; Gibco) supplemented with 20% fetal bovine serum (#26140–079; Gibco, Grand Island, NY), HAT supplement (100 μM sodium hypoxanthine, 400 nM aminopterin, 16 μM thymidine; #21060–017; Gibco), and penicillin/streptomycin (#17-602E, Lonza, Switzerland). F-11 cells were tested for mycoplasma contamination using Universal Mycoplasma Detection Kit (# ATCC 30–1012K, ATCC, Manassas, VA) and found to be free of contamination. F-11 cells for imaging experiments were plated on Poly-Lysine (#P1274, Sigma, St. Louis, MO) coated 0.15 mm x 25 mm coverslips (#64–0715 (CS-25R15), Warner Instruments, Hamden, CT) in a six-well plate. Cells were transfected with Lipofectamine 2000 (4 μl/well, Invitrogen, Grand Island, NY) reagent using 1–3 μg of cDNA per well. 24 hr post-transfection, media was replaced with HEPES-buffered saline (HBR, double deionized water and in mM: 140 NaCl, 4 KCl, 1 $MgCl_2$, 1.8 $CaCl_2$, 10 HEPES (free acid) and five glucose) for at least 2 hr prior to the imaging. During experiments, cells were treated with 100 ng/ml NGF 2.5S (#13257–019, Sigma), vehicle (HBR) or 20 nM wortmannin (Sigma W1628).

TRPV1-cYFP (rat) (*Ufret-Vincenty et al., 2015*), TRPV1-ARD-ctagRFP (rat), TRPV2-cYFP (rat) (*Mercado et al., 2010*) DNA constructs were made in the pcDNA3 vector (Invitrogen), where '-n' or '-c' indicates that the fluorescent protein is on the N- or C-terminus, respectively. TRPV4-EGFP (human) in pEGFP was obtained from Dr. Tim Plant (Charite-Universitatsmedizine, Berlin) (*Strotmann et al., 2003*). TrkA (rat) in the pcCMV5 vector and p75NTR (rat) in the pcDNA3 vector were obtained from Dr. Mark Bothwell (University of Washington, Seattle). PH-Akt-cCerulean in the pcDNA3-k vector was made based on the construct in the pHR vector from Dr. Orion Weiner's Lab (*Toettcher et al., 2011*). The function of the ion channels tested were confirmed using $Ca^{2+}$ imaging and/or patch clamp electrophysiology (data not shown).

## Western blotting

For detection of relative expression of PI3K p85α subunit, cells were transfected as described above for imaging experiments. 24 hr after transfection, cells were scraped off the bottom of 10 cm plates, washed with Phosphate Buffered Saline (PBS) 4 times and homogenized in Lysis buffer (1% Triton 25 mM Tris-HCl, 150 mM NaCl, 1 mM EDTA, pH 7.4) for 2 hr with mixing at 4°C. Lysates were spun down at 14000 rpm for 30 min at 4°C to remove the cell nuclei and debris. Cleared lysates were mixed with Laemmli 2x SDS sample buffer (#161–0737, Bio-Rad, Hercules, CA), boiled for 10 min and subjected to SDS PAGE to separate proteins by size. Gels were then transferred onto the PDVF membrane using Trans-Blot SD semi-dry transfer cell (Bio-Rad) at 15 V for 50 min. Membranes were blocked in 5% BSA Tris-Buffered Saline, 0.1% Tween (TBS-T) for 1 hr and probed with primary antibody for 1 hr at RT. Next, membranes were washed 6x times with TBS-T and probed with secondary antibodies conjugated with Horse Radish Peroxidase (HRP) for 1 hr. After another set of 6 washes membranes were developed by addition of the SuperSignal West Femto HRP substrate (#34096, Thermo, Grand Island, NY) and imaged using CCD camera-enabled imager. For quantification, blot images were analyzed in ImageJ. ROIs of the same size were drawn around the bands for p85 and tubulin, then mean pixel intensity was measured. Mean p85 intensities were normalized by dividing by mean tubulin intensities and plotted in *Figure 2—figure supplement 3*. Experiments were repeated with n = 5 independent samples. Primary antibodies used were: anti-PI3K (alpha) polyclonal (#06–497 (newer Cat#ABS234), Upstate/Millipore, Burlington, MA) at 1:600 dilution; β Tubulin (G-8) (#sc-55529, Santa Cruz, Dallas, TX) at 1:200 dilution. Secondary antibodies used: Anti-Rabbit

IgG (#074–1506, KPL/SeraCare Life Sciences, Milford, MA) at 1:30,000 dilution; Anti-Mouse IgG (#NA931, Amersham/GE Healthcare Life Sciences, United Kingdom) at 1:30,000 dilution.

For detection of phosphorylated Akt, cells plated in six-well plates were treated for the indicated amount of time (*Figure 3*, *Figure 1—figure supplement 2*) in the $CO_2$ incubator at 37°C. Immediately after treatment, wells were aspirated and scraped in ice-cold lysis buffer ($H_2O$, TBS, 1% NP-40, 5 mM NaF, 5 mM $Na_3VO_4$ with added Protease inhibitors (#P8340, Sigma) and Phosphatase Inhibitor Cocktail 2 (#P5726, Sigma). After incubation on ice for 15 min, lysates were cleared by centrifugation at 15 k g for 15 min at 4°C. Protein contents of cleared lysates were measured using the BCA assay (#23225 Pierce) according to manufacturer's protocol. Volumes of lysates were adjusted according to these measurements and subjected to SDS-PAGE. Gels were transferred onto PVDF membranes using wet-transfer. Membranes were blocked in TBS-T with 5% milk for 1 hr and incubated overnight at 4°C with one of the following primary antibodies: pAKTs473 clone D9E (#4060, Cell Signaling), pAKTt308 clone 244F9 (#4056, Cell Signaling) . Further procedures were as indicated in the previous paragraph. After development membranes were stripped using Pierce Restore Western Blot stripping buffer (#21059, Thermo Fisher), reprobed with the other anti-phospho-AKT antibody and then stripped and re-probed with panAKT clone 40D4 (#2920, Cell Signaling) antibody at 1:2500 dilution. Data was normalized by diving the average intensity of a band by the average intensity of a blot and then dividing by that of a pan-Akt blot (*Figure 3*).

## Acknowledgements

We thank Mika Munari, Gilbert Martinez, Mark Bothwell, Bertil Hille, William Zagotta, Shao-En Ong, Tamara Rosenbaum and Gaby Bergollo for helpful discussions. We are grateful to the following individuals for providing cDNA constructs: Dr. Tim Plant (Charite-Universitatsmedizine, Berlin) for TRPV4; Dr. Mark Bothwell (University of Washington, Seattle) for TrkA and p75$_{NTR}$; and Dr. Orion Weiner (UCSF) for PH-Akt.

Research reported in this publication was supported by the National Eye Institute of the National Institutes of Health under award numbers R01EY017564 (to SEG), by the National Institute of General Medical Sciences of the National Institutes of Health under award numbers R01GM100718 and R01GM125351 (to SEG), by the National Institute of Mental Health under award number R01MH113545 (to SEPS), by the National Institute of Biomedical Imaging and Bioengineering of the National Institutes of Health under award number T32EB001650 (to AS), by the following additional awards from the National Institutes of Health: S10RR025429, P30DK017047, and P30EY001730. and by a Royalty Research Fund Award from the University of Washington (to SEG). The content is solely the responsibility of the authors and does not necessarily represent the official views of the National Institutes of Health. The authors declare no competing financial interests.

## Additional information

### Funding

| Funder | Grant reference number | Author |
| --- | --- | --- |
| National Eye Institute | R01EY017564 | Sharona E Gordon |
| National Institute of General Medical Sciences | R01GM100718 | Sharona E Gordon |
| National Institute of General Medical Sciences | R01GM125351 | Sharona E Gordon |
| University of Washington | Royalty Research Fund | Sharona E Gordon |
| National Institute of Mental Health | R01MH113545 | Stephen EP Smith |
| National Institute of Biomedical Imaging and Bioengineering | T32EB001650 | Anastasiia Stratiievska |
| National Institutes of Health | S10RR025429 | Sharona E Gordon |
| National Institutes of Health | P30DK017047 | Sharona E Gordon |

| National Institutes of Health | P30EY001730 | Sharona E Gordon |

The funders had no role in study design, data collection and interpretation, or the decision to submit the work for publication.

## Author contributions

Anastasiia Stratiievska, Conceptualization, Formal analysis, Investigation, Methodology, Writing—original draft, Writing—review and editing; Sara Nelson, Investigation, Methodology; Eric N Senning, Stephen EP Smith, Conceptualization, Writing—review and editing; Jonathan D Lautz, Investigation, Methodology, Writing—review and editing; Sharona E Gordon, Conceptualization, Formal analysis, Funding acquisition, Investigation, Methodology, Writing—original draft, Writing—review and editing

## Author ORCIDs

Anastasiia Stratiievska (iD) http://orcid.org/0000-0002-5523-0773
Sharona E Gordon (iD) http://orcid.org/0000-0002-0914-3361

## Decision letter and Author response

Decision letter https://doi.org/10.7554/eLife.38869.023
Author response https://doi.org/10.7554/eLife.38869.024

## Additional files

### Supplementary files

• Source data 1. Source data from figures. Excel file containing source data from the figures as indicated. The name of Excel sheet corresponds to the figure to which it is related
DOI: https://doi.org/10.7554/eLife.38869.019

• Transparent reporting form
DOI: https://doi.org/10.7554/eLife.38869.021

### Data availability

All data generated or analysed during this study are included in the manuscript and supporting files.

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
