## [Decision Letter]

[Editors’ note: this article was originally rejected after discussions between the reviewers, but the authors were invited to resubmit after an appeal against the decision.]

Thank you for submitting your work entitled "Reciprocal regulation among TRPV1 channels and phosphoinositide 3-kinase in response to nerve growth factor" for consideration by *eLife*. Your article has been reviewed by three peer reviewers, and the evaluation has been overseen by a Reviewing Editor and a Senior Editor. The reviewer #1 (Tibor Rohacs) agreed to be identified.

Our decision has been reached after consultation between the reviewers. Based on these discussions and the individual reviews below, we regret to inform you that your work will not be considered further for publication in *eLife*.

Your study shows for the first time that TRP channels can potentiate the activity of PI3 kinase and thereby upregulate their trafficking to the membrane via PIP_2_ or PIP_3_. In contrast to chanzymes where the enzyme is fused to the channel, this study highlights an alternate pathway for modulation of enzyme activity by TRP channels. Nevertheless, as detailed below, the reviewers have noted several concerns that raise questions about the mechanism of modulation. The consensus is that these concerns cannot be addressed in a two-month timeframe.

Briefly:

1) Without demonstrating that the specific mutations on TRPV1 can abrogate PI3K potentiation, it is difficult to make the case that TRPV1 directly modulates PI3K activity.

2) Related to the above point, all the assays showing potentiation are in vivo assays. This does not rule out the possibility that this modulation is indirect or is mediated by other partners. The real test is to demonstrate that AKD directly modulates PI3K activity in vitro. This will also rule out the possibility that TRPV1 is inhibiting lipid phosphatases rather than potentiating PI3Kinase.

3) Given that TRPM4 which does not have ARDs also potentiates PI3K significantly without increasing the trafficking of these channels to the membrane, one wonders whether ARD is essential for potentiation. This means that the ARD is not the only component responsible for potentiation and it is not clear to what extent ARDs contribute to potentiation compared to other structural elements in TRP channels.

Reviewer #1:

The manuscript by Stratiievska et al. is based on an intriguing and unexpected finding: the presence of TRPV1 increases the activity of PI3K. The effect is robust, and several other TRP channels with ankyrin repeat domains (ARD) exert a similar effect, even though the effects size is smaller than with TRPV1. The isolated ARD of TRPV1 also increases the activity of PI3K, even though the effect on kinetics is far less pronounced than that by the full length TRPV1. The authors also show that Akt-phosphorylation is increased by TRPV1, which provides an independent verification of increased PI3K activity. The data can be placed in a signaling paradigm where TRPV1 potentiates the NGF-induced activity of PI3K, leading to higher PI(3,4,5)P3 levels, which stimulates TRPV1 trafficking to the plasma membrane.

In this reviewer's opinion the data are intriguing, novel, and unexpected, and as such they will potentially be influential and may even lead to paradigm shift(s) in our thinking. To strengthen the paper, I recommend some additional control experiments, and also to consider alternative possibilities in data interpretation.

1) The authors present the Akt-PH-GFP as a selective PI(3,4,5)P3 sensor. This construct however binds to both PI(3,4)P2 and PI(3,4,5)P3, see Balla et al. TIPS 2000 (PMID: 10871889) for references. This by itself is not a major problem, as both these lipids are products of PI3K; nevertheless, I recommend that the authors cite it as a PI(3,4,5)P3 / PI(3,4)P sensor. It would also be nice to confirm the key finding, that is increased PI(3,4,5)P3 production in the presence of TRPV1, using a more specific PI(3,4,5)P3 sensor, for example the Btk-PH-GFP.

2) Another simple control experiment that would strengthen the paper, is to inhibit PI3K with either wortmannin or LY29004 and demonstrate that the translocation is inhibited both in TRPV1 expressing and non-expressing cells.

3) The authors mention in their Abstract: "Further, other TRPV channels with conserved ARDs also potentiated NGF‐induced PI3K activity whereas TRP channels lacking ARDs did not." This statement is at odds with Figure 5—figure supplement 2, where TRPM4 which has no ARD-s is shown to also potentiate PI3K activity, and this effect is similar in size to that evoked by TRPV2 and TRPV4, but much smaller than that induced by TRPV1. While the authors provide evidence that the TRPV1 ARD is sufficient to potentiate PI3K, there is no such evidence provided for other TRP-s and I do not think the data makes a case that ARDS-s potentiate PI3K in general.

4) The authors claim throughout the paper and in the Abstract and in the title that PI3K activity is increased in the presence of TRPV1. This is quite likely but there is an alternative explanation: what if TRPV1 inhibits the activity of lipid phosphatases that break down PI(3,4,5)P3 and PI(3,4)P2. The findings would be the same. If the authors can come up with an experiment to test this, it would strengthen the manuscript. If not, this possibility needs to be discussed, and the conclusions should be stated more cautiously.

Reviewer #2:

TRP channels are subject of cellular regulation at different levels. While the polimodal activation mechanism allow them to serve as coincidence detectors for cellular sensing, sensitization and desensitization of different kind contribute to the fine-tuning of their cellular activity. Important for the background of the present manuscript, trafficking in and out of the plasma membrane provides further dynamic control of TRP's current density.

First reported by the Clapham laboratory, several groups have studied TRP channel trafficking during the last 10 years, including the Gordon's laboratory. It is now accepted that part of NGF-dependent sensitization is caused by traffic and that IP3K pathway is part of this mechanism. This present work, which is a follow up of Dr. Gordon's past works on TRPV1 trafficking, is based mainly on observations performed in live cell imaging recordings under TIRF configuration. The authors identify two phenomena, first that TRPV1 traffic to the plasma membrane is associated to an increase in PIP_3_ and secondly that the activity of PI3K is somewhat associated to the expression levels of TRPV1 channels. In brief, the manuscript sketch two stories, both undone.

There is a substantial amount of literature on this topic making the first story – the one about PIP_3_ levels and TRPV1 trafficking- definitely trivial and the conclusions not original enough for the scope of this journal. On the other hand, the second story, dealing with the ARD-dependent regulation of IP3K (underscored at the title of the manuscript) is certainly of potential interest, but needs additional data to support the conclusions drawn by the authors.

In general the manuscript tends to be repetitive, in both the text and figures, a sharper text leading to the conclusions would be appreciated.

Several points in the manuscript need to be addressed.

Specific comments:

From Stein et al., 2006 and Zhang, Huang and McNaughton et al., 2005, we know that a) TRPV1 binds to PI3K and b) inhibition of the latter diminish the number of NGF responsive cells. As the authors state in the last paragraph of the Introduction, "whether TRPV1/p85 contributes to NGF-induced trafficking of TRPV1 is unknown", however, no attempt to address this deeply is observed. The authors showed a correlation between TRPV1 trafficking and increased levels of PIP_3_ (indirectly by means of an Akt-PH fluorescent probe). Together with this, they showed that the expression of TRPV1 ARD is sufficient to cause and increase in Akt-PH signal at the membrane.

1) The question remains, is it the increase in PIP_3_ levels or the levels of PIP_2_, generated also by PI3K activity, the signal leading to the increased traffic of TRPV1? Experiments aiming to solve this would certainly improve the trafficking section.

2) Does PI3K binds to TRPV1 ARD? A coIP is needed to show evidence of such interaction.

3) In Figure 2 the authors show differences in the time course of Akt-PH fluorescence and TRPV1 fluorescence at the plasma membrane. Is it that PIP_3_-Akt-PH gets internalized? Is it that PI3K detaches from TRPV1 ARD (in case there is direct interaction) leading to a lower levels of localized enzyme at the membrane? Probably live cell imaging colocalization between TRPV1 and PI3K might shed light into this.

4) In the last paragraph of the subsection “NGF induces production of PIP_3_ by P13K followed by trafficking of TROV1 channels to the PM” authors claim that Fngf of 1.5 represent about ten-fold response. Such difference can be easily observed in a simple biotinylation experiment and will serve to confirm and calibrate the imaging values.

5) The authors state that other TRPs seems to respond in a similar fashion TRPV1 does, although the data doesn't seem to support such strong claim. The differences are not significant enough or the response is modest. Probably the authors should lower the tone of their statements throughout the manuscript.

6) The information that can be extracted from ARD/PI3K interaction data set is limited. Why the Akt-PH signal increases at the plasma membrane when the overexpressed ARD is soluble?

7) Any soluble ARD can do the trick of potentiating PI3K after NGF incubations? TRP channels ARD have special features absent in other canonical ARDs, are these helping to the phenotype observed?

8) Can the soluble ARD from TRPV1 induce a higher traffic on the other TRPs having a modest response to NGF?

9) The authors assume that the putative interaction between PI3K and TRPV1 ARD is somewhat associated to allosteric regulation of PI3K activity without any other proof but indirect measurements of PIP_3_ levels. Again, would be desirable to observe such interaction in a biochemistry assay and make sure that the ARD is not acting just as scaffold for additional proteins modulating PI3K activity.

Methods and statistics

10) While this reviewer recognizes the efforts made on image analysis, the authors are working under assumptions that are out of their control. First, all the numerical analysis based on Mattheyses and Axelrod 2006 assume a critical angle they can't measure. For that reason in the cited article, the authors used beads of known size to calibrate the evanescent field. Moreover, to define layers they are assuming the single emitters are of the same size and intensity. Probably single emitters are having the same intensity, however multiple emitters can be together in a vesicle or membrane cluster or groups of vesicles in a non-predictable fashion. Therefore the definition of the different layers is not as clear as it seems from the Materials and methods section.

11) The authors claim that the population doesn't distribute normal within themselves (subsection “NGF induces production of PIP_3_ by PI3K followed by trafficking of TRPV1 channels to the PM”), however they used a parametric test for paired data that assumes normal distribution. Moreover, how the authors deal with outliers? Potential outliers are visible in Figure 2D, Figure 3C and Supplementary Figure 5B.

Reviewer #3:

In a previous work from this group (Stein et al., 2006), they showed that PI3K directly interacts with the N-terminal region of TRPV1 (residues 1 to 432) and that NGF increases the number of channels in the plasma membrane. Based on these results, this group proposed a model for NGF-mediated hyperalgesia in which PI3K facilitates trafficking of TRPV1 to the plasma membrane. In the present study, the authors used total internal reflection fluorescence microscopy to show that TRPV1 potentiates NGF-induced PI3K activity and that its ankyrin repeats (residues 111 to 359) were sufficient to produce this potentiation. Basically, the previous proposed mechanism has not changed in that NGF-induced PI3K activity promotes channel trafficking; however, the new data show that in the presence of TRPV1 the PI3K activity is enhanced as the PIP_3_ levels increase in the plasma membrane. In addition, the authors narrowed down the interaction site to ~ 248 residues at the ankyrin repeats domain (ARD). The authors followed a rational plan and performed well-thought experiments to determine that TRPV1 potentiates the activity of PI3K.

1) The authors should determine the TRPV1-PI3K interaction site (currently located within ~248 residues of the ARD), since they have a good readout with the TRPV1-ARD experiments (as shown in Figure 5B). I suggest performing a sequence alignment and analyze the conserved residues between channels; this analysis can be used to generate new TRPV1-ARD constructs that might lack the potentiation effect and determine the TRPV1-PI3K interaction site. This is important since targeting this site could help modulate the NGF-TRPV1 mediated sensitization.

2) The authors should repeat the experiment shown in Figure 6B using a mammalian TRPA1 instead of the zebrafish one, as these channels would likely display higher expression levels in F11 cells. This would help determining whether TRPA1 significantly potentiates NGF‐induced PI3K activity.

3) In Figure 5—figure supplement 2A, the authors should increase the number of samples for TRPV4 (equivalent to the other channels). In its present form, most of the data points lie within the distribution of the control.

4) The authors should include in Figure 5C the data points corresponding to the control and TRPV1.

5) Although the experiments with TRPM4 and TRPM8 show that they do not have reciprocal regulation, both channels display opposite effects for PI3K activity and channel trafficking (Figures 6B and 6D). The authors should provide a deeper discussion for this opposite effect.

6) Testing channels (TRPM) without ARD is a good idea; however, coming back to suggestion #1, I think that finding a TRPV1 and/or TRPV1-ARD construct that decouples potentiation of PI3K activity from the expression will support the model proposed in Figure 7.

[Editors’ note: what now follows is the decision letter after the authors submitted for further consideration.]

Thank you for submitting your article "Reciprocal regulation among TRPV1 channels and phosphoinositide 3-kinase in response to nerve growth factor" for consideration by *eLife*. Your article has been reviewed by two peer reviewers, and the evaluation has been overseen by a Reviewing Editor and Richard Aldrich as the Senior Editor. The following individual involved in review of your submission has agreed to reveal his identity: Tibor Rohacs (Reviewer #1).

The reviewers have discussed the reviews with one another and the Reviewing Editor has drafted this decision to help you prepare a revised submission.

Summary:

In this short report, Stratiievska et al. show for the first time that TRP channels can potentiate the activity of PI3 kinase and thereby upregulate their trafficking to the membrane via PIP_2_ or PIP_3_. In contrast to chanzymes where the enzyme is fused to the channel, this study highlights an alternate pathway for modulation of enzyme activity by TRP channels. While the reviewers were pleased with the trimmed version of the manuscript, they are not satisfied with the revised version. Specifically, they have asked for the following controls:

Essential revisions:

1) The Akt-PH domain is a combined PI(3,4,5)P3 and PI(3,4)P2 sensor. The authors argued that their Akt phosphorylation assay is an independent readout of PI(3,4,5)P3 production; therefore there is no need to repeat the experiment with a more specific PI(3,4,5)P3 sensor. I disagree with this argument. My understanding of Akt regulation is that this enzyme binds to the plasma membrane via its PH domain that binds both PI(3,4,5)P3 and PI(3,4)P2, and as a result it becomes phosphorylated by PDK1, see Balla Physiological Reviews 2013. See also Franke et al. Science 1997 PMID:9005852, showing that PI(3,4)P2 stimulates Akt. In other words, Akt phosphorylation is also a combined PI(3,4,5)P3 PI(3,4)P2 assay.

As the data stands, everything can be explained by a potentiated increase in PI(3,4)P2, which is also increased upon PI3K stimulation. This is not a very likely scenario of course, but it is very easy to test. I strongly recommend that the authors perform the experiment with a more specific PI(3,4,5)P3 sensor.

2) My original request was to test the effect of wortmannin on the translocation of the Akt-PH domain, not that of TRPV1. Again, this is an extremely simple experiment, and would solidify the key finding.

3) The authors forgot to delete the second half of the following sentence from the Abstract: "Further, other TRPV channels with conserved ARDs also potentiated NGF‐induced PI3K activity whereas TRP channels lacking ARDs did not." Also the effects of TRPV4 and TRPV2 on Akt-PH translocation look smaller than that induced by TRPV1, are the effects significantly different from TRPV1? I recommend that this is tested, and shown in Table 1 and mentioned in the Results.

---

## [Author Response]

[Editors’ note: the author responses to the first round of peer review follow.]

Your study shows for the first time that TRP channels can potentiate the activity of PI3 kinase and thereby upregulate their trafficking to the membrane via PIP2 or PIP3. In contrast to chanzymes where the enzyme is fused to the channel, this study highlights an alternate pathway for modulation of enzyme activity by TRP channels. Nevertheless, as detailed below, the reviewers have noted several concerns that raise questions about the mechanism of modulation. The consensus is that these concerns cannot be addressed in a two-month timeframe. Briefly:1) Without demonstrating that the specific mutations on TRPV1 can abrogate PI3K potentiation, it is difficult to make the case that TRPV1 directly modulates PI3K activity.

In response to the reviewers’ and Editor’s comments, we have narrowed the focus of the manuscript, stressing its most important and most robust findings. Thus, we have changed the format to that of a Short Report, which better reflects the high significance of our data and is more appropriate given the scope of the mechanism elucidated.

The present work demonstrates a previously unknown potentiation of NGF-induced PI3K activity by TRPV1. The importance of this form of regulation cannot be understated, as PI3K regulation of TRPV1 during inflammation is a well-established pathway for peripheral sensitization to painful stimuli. We show definitively that the ARD domain of TRPV1 is sufficient for its regulation of PI3K. Does the regulation involve a direct interaction between the ARD domain and PI3K? Our work does not rule out that adapter proteins or additional regulatory proteins may be involved. However, the novelty and significance of the reciprocal regulation we discovered stand on their own, even without a full understanding of the details of the regulation and the many downstream steps that are involved in enhanced channel trafficking. The reciprocal regulation of TRPV1 and PI3K, together with sufficiency of the ARD domain of TRPV1 in supporting the reciprocal regulation with PI3K, make our work timely, compelling, and of broad interest.

There is good reason to believe that the allosteric regulation of PI3K by TRPV1 is direct: we previously showed that the N-terminal domain of TRPV1 interacts directly with PI3K using recombinant protein fragments, cell pull-downs, and yeast 2-hybrid assays. We agree with the Reviewer, though, that the regulation shown here need not be direct. Indeed, even mutations that disrupt the PI3K-TRPV1 interactions and the reciprocal regulation would not rule out a role for additional adapter/regulatory proteins. We are now more circumspect in our framing of the mechanism of the reciprocal regulation, and take care to discuss the possibility that it may not involve a direct interaction.

2) Related to the above point, all the assays showing potentiation are in vivo assays. This does not rule out the possibility that this modulation is indirect or is mediated by other partners. The real test is to demonstrate that AKD directly modulates PI3K activity in vitro. This will also rule out the possibility that TRPV1 is inhibiting lipid phosphatases rather than potentiating PI3Kinase.

We agree that a cell-free in vitro assay is needed to rule out a role for other cellular components in mediating the potentiation of PI3K by TRPV1. Unfortunately, such an essay is not standard in the literature and would need to be developed. The activation mechanism of PI3K involves relief of autoinhibition through binding of PI3K to phosphorylated tyrosines on a cell-surface receptor. Its recruitment to the membrane is also required. These experiments can therefore not be performed using soluble proteins in a test tube. To our knowledge, only one lab in the world has an appropriate biochemical assay using a supported bilayer. We have established a collaboration with this lab, but significant additional development of the assay will be required to allow it to be used along the lines suggested by the Editor and reviewers. As it stands now, there is no assay that can be used to address the question in vitro. As discussed above, we have reorganized and shortened the manuscript to focus on the sufficiency of the ARD domain of TRPV1 to support the NGF-induced potentiation of PI3K. We hope our shift of focus from the direct nature of the interaction to the sufficiency of the interaction in a cell environment will satisfy the Editor and reviewers.

3) Given that TRPM4 which does not have ARDs also potentiates PI3K significantly without increasing the trafficking of these channels to the membrane, one wonders whether ARD is essential for potentiation. This means that the ARD is not the only component responsible for potentiation and it is not clear to what extent ARDs contribute to potentiation compared to other structural elements in TRP channels.

We agree with the Editor and reviewers that our work does not address the possibility that other pathways of potentiation may well be present in cells. We have therefore more tightly focused the manuscript on the TRPV channels, the one potentiation pathway about which we have information. Focusing on TRPV channels only allows us to make mechanistic conclusions about the sufficiency of the TRPV ARD in potentiation, even though we cannot argue that it is the exclusive pathway by which PI3K may be regulated.

Reviewer #1:[…] To strengthen the paper, I recommend some additional control experiments, and also to consider alternative possibilities in data interpretation.1) The authors present the Akt-PH-GFP as a selective PI(3,4,5)P3 sensor. This construct however binds to both PI(3,4)P2 and PI(3,4,5)P3, see Balla et al. TIPS 2000 (PMID: 10871889) for references. This by itself is not a major problem, as both these lipids are products of PI3K; nevertheless, I recommend that the authors cite it as a PI(3,4,5)P3 / PI(3,4)P sensor. It would also be nice to confirm the key finding, i.e. increased PI(3,4,5)P3 production in the presence of TRPV1, using a more specific PI(3,4,5)P3 sensor, for example the Btk-PH-GFP.

The suggested citation has been added and we have modified the text to reflect the mixed specificity of Akt-PH. Our analysis of PIP_3_-dependent phosphorylation of the Akt enzyme provides orthogonal evidence that TRPV1 potentiates NGF-induced PI3K activity. In our opinion, this sort of orthogonal readout is better in confirming the enhanced PIP_3_ generation in TRPV1-expressing cells than using another PH probe would be.

2) Another simple control experiment that would strengthen the paper, is to inhibit PI3K with either wortmannin or LY29004 and demonstrate that the translocation is inhibited both in TRPV1 expressing and non-expressing cells.

In a previous publication, we showed than wortmannin prevents NGF-induced trafficking of TRPV1 to the plasma membrane. We now cite this work more prominently.

3) The authors mention in their Abstract: "Further, other TRPV channels with conserved ARDs also potentiated NGF‐induced PI3K activity whereas TRP channels lacking ARDs did not." This statement is at odds with Figure 5—figure supplement 2, where TRPM4 which has no ARD-s is shown to also potentiate PI3K activity, and this effect is similar in size to that evoked by TRPV2 and TRPV4, but much smaller than that induced by TRPV1. While the authors provide evidence that the TRPV1 ARD is sufficient to potentiate PI3K, there is no such evidence provided for other TRP-s and I do not think the data makes a case that ARDS-s potentiate PI3K in general.

As discussed in response to Editor’s comment #1, we have narrowed the focus of our paper to TRPV channels. We hope the reviewer views this change, along with the increased emphasis on the sufficiency of the ARD from TRPV1, as sufficient to address this concern.

4) The authors claim throughout the paper and in the Abstract and in the title that PI3K activity is increased in the presence of TRPV1. This is quite likely but there is an alternative explanation: what if TRPV1 inhibits the activity of lipid phosphatases that break down PI(3,4,5)P3 and PI(3,4)P2. The findings would be the same. If the authors can come up with an experiment to test this, it would strengthen the manuscript. If not, this possibility needs to be discussed, and the conclusions should be stated more cautiously.

We agree with the reviewer that a more complicated model, in which TRPV1 binds to PI3K yet also exerts an effect on its cognate phosphatase, is unlikely. We nonetheless added a discussion of this alternative explanation to the text.

Reviewer #2:TRP channels are subject of cellular regulation at different levels. While the polimodal activation mechanism allow them to serve as coincidence detectors for cellular sensing, sensitization and desensitization of different kind contribute to the fine-tuning of their cellular activity. Important for the background of the present manuscript, trafficking in and out of the plasma membrane provides further dynamic control of TRP's current density.First reported by the Clapham laboratory, several groups have studied TRP channel trafficking during the last 10 years, including the Gordon's laboratory. It is now accepted that part of NGF-dependent sensitization is caused by traffic and that IP3K pathway is part of this mechanism. This present work, which is a follow up of Dr. Gordon's past works on TRPV1 trafficking, is based mainly on observations performed in live cell imaging recordings under TIRF configuration. The authors identify two phenomena, first that TRPV1 traffic to the plasma membrane is associated to an increase in PIP3 and secondly that the activity of PI3K is somewhat associated to the expression levels of TRPV1 channels. In brief, the manuscript sketch two stories, both undone.There is a substantial amount of literature on this topic making the first story – the one about PIP3 levels and TRPV1 trafficking- definitely trivial and the conclusions not original enough for the scope of this journal. On the other hand, the second story, dealing with the ARD-dependent regulation of IP3K (underscored at the title of the manuscript) is certainly of potential interest, but needs additional data to support the conclusions drawn by the authors.In general the manuscript tends to be repetitive, in both the text and figures, a sharper text leading to the conclusions would be appreciated.Several points in the manuscript need to be addressed.Specific comments:From Stein et al., 2006 and Zhang, Huang and McNaughton et al., 2005, we know that a) TRPV1 binds to PI3K and b) inhibition of the latter diminish the number of NGF responsive cells. As the authors state in the last paragraph of the Introduction, "whether TRPV1/p85 contributes to NGF-induced trafficking of TRPV1 is unknown", however, no attempt to address this deeply is observed. The authors showed a correlation between TRPV1 trafficking and increased levels of PIP_3_ (indirectly by means of an Akt-PH fluorescent probe). Together with this, they showed that the expression of TRPV1 ARD is sufficient to cause and increase in Akt-PH signal at the membrane.1) The question remains, is it the increase in PIP3 levels or the levels of PIP2, generated also by PI3K activity, the signal leading to the increased traffic of TRPV1? Experiments aiming to solve this would certainly improve the trafficking section.

As the reviewer notes, the present paper is about potentiation of NGF-induced PI3K activity, and enhanced levels of PIP_3_, that occur in TRPV1-expressing cells and not on the downstream signaling by which increases in PIP_3_ lead to insertion of TRPV1 channels into the plasma membrane. The framework for our Discussion, however, was clearly insufficient in making the scope of the manuscript clear. The improved, narrower focus more clearly shows that the scope of the present manuscript includes the effect of TRPV1 expression on PI3K activity but not downstream signaling and channel trafficking.

We are puzzled by the reviewer’s comment that PI3K signaling will increase the levels of PIP_2_. To our knowledge this has not been reported, and indeed would be unexpected. We hope that narrowing the focus of the work to TRPV channels only, together with combining figures to highlight the importance of the ARD in mediating PI3K potentiation, will more clearly signal the manuscript’s topic.

2) Does PI3K binds to TRPV1 ARD? A coIP is needed to show evidence of such interaction.

We have previously shown that the p85 subunit of PI3K binds directly to the N-terminal region of TRPV1, which includes the ARDs. The interaction was shown to be direct in our previous publication using a binding assay with a recombinant fragment. Although narrowing down the region of the ARDs that is involved is a priority, as is localizing the ARD binding site on PI3K, it is beyond the scope of the present work.

3) In Figure 2 the authors show differences in the time course of Akt-PH fluorescence and TRPV1 fluorescence at the plasma membrane. Is it that PIP_3_-Akt-PH gets internalized? Is it that PI3K detaches from TRPV1 ARD (in case there is direct interaction) leading to a lower levels of localized enzyme at the membrane? Probably live cell imaging colocalization between TRPV1 and PI3K might shed light into this.

The kinetics of the Akt-PH probe translocation are governed in part by the probe’s affinity for PIP_3_. Therefore, the differences in the time courses of probe translocation and channel trafficking are not meaningful. If we had used a higher affinity probe, we would have seen a faster response. It should be clarified that Akt-PH is a soluble cytoplasmic protein which is not internalized in the same manner as transmembrane proteins. In addition, the Akt-PH probe detects the lipid product, which is generated at an early stage of the signaling pathway leading to the channel trafficking.

Colocalization of PI3K with TRPV1 might indeed be useful. However, we and others find that overexpression of PI3K in cells produces profoundly unhealthy cells, as excessive PIP_3_ is toxic. Thus, the suggested co-localization studies are not presently possible.

4) In the last paragraph of the subsection “NGF induces production of PIP_3_ by P13K followed by trafficking of TROV1 channels to the PM” authors claim that Fngf of 1.5 represent about ten-fold response. Such difference can be easily observed in a simple biotinylation experiment and will serve to confirm and calibrate the imaging values.

This would be a useful calibration, but is not feasible due to the details of TRPV1 expression. In F-11 cells and DRG cells, nearly all TRPV1 is present in intracellular membranes, presumably the ER. We estimate that well under 1% of TRPV1 is localized to the surface. Thus, even if only a small fraction of cells are permeable to the biotinylation reagent due to damage/death, the off-surface labeling dwarfs the true surface labeling. Perhaps more importantly, we give the estimate of ten-fold only to illustrate the gross underestimate of change in signal that occurs in TIRF microscopy. We now state more clearly that this is only a rough estimate referring to the order of magnitude expected.

5) The authors state that other TRPs seems to respond in a similar fashion TRPV1 does, although the data doesn't seem to support such strong claim. The differences are not significant enough or the response is modest. Probably the authors should lower the tone of their statements throughout the manuscript.

Please see responses to Editor point #1 and reviewer #1 point #3.

6) The information that can be extracted from ARD/PI3K interaction data set is limited. Why the Akt-PH signal increases at the plasma membrane when the overexpressed ARD is soluble?

We hypothesize that the soluble ARD interacts with cytosolic PI3K in resting cells. Upon stimulation with NGF, the ARD/PI3K complex would be recruited to the membrane, PI3K would be activated, and PIP_3_ would be generated. The newly generated PIP_3_ would then bind Akt-PH and result in an increase in Akt-PH associated fluorescence near the membrane. The narrower focus of the revised manuscript now states this model more clearly.

7) Any soluble ARD can do the trick of potentiating PI3K after NGF incubations? TRP channels ARD have special features absent in other canonical ARDs, are these helping to the phenotype observed?

We are eager to learn the answer to this question and will address it in future work.

8) Can the soluble ARD from TRPV1 induce a higher traffic on the other TRPs having a modest response to NGF?

This is a great suggestion. Although beyond the scope of the present manuscript, we will incorporate the experiment into subsequent work.

9) The authors assume that the putative interaction between PI3K and TRPV1 ARD is somewhat associated to allosteric regulation of PI3K activity without any other proof but indirect measurements of PIP3 levels. Again, would be desirable to observe such interaction in a biochemistry assay and make sure that the ARD is not acting just as scaffold for additional proteins modulating PI3K activity.

In addition to the experiments to which the reviewer refers, in which fluorescently labeled PIP_3_-binding domains were used to indirectly measure PIP_3_ levels in the plasma membrane, we used an orthogonal assay based on phosphorylation of the kinase Akt at two positions. The data from these experiments, shown in Figure 3, fully support the interpretation of our imaging data that TRPV1 expression leads to enhanced levels of PIP_3_ in response to nerve growth factor. Please also see response to Editor point #2.

Methods and statistics10) While this reviewer recognizes the efforts made on image analysis, the authors are working under assumptions that are out of their control. First, all the numerical analysis based on Mattheyses and Axelrod 2006 assume a critical angle they can't measure. For that reason in the cited article, the authors used beads of known size to calibrate the evanescent field. Moreover, to define layers they are assuming the single emitters are of the same size and intensity. Probably single emitters are having the same intensity, however multiple emitters can be together in a vesicle or membrane cluster or groups of vesicles in a non-predictable fashion. Therefore the definition of the different layers is not as clear as it seems from the Materials and methods section.

The model described in the Materials and methods is used strictly to demonstrate that the measured change in PIP_3_ probe intensity underestimates the actual change in PIP_3_. No data analysis was based on this model, and we now more clearly label it a “rough” estimate.

11) The authors claim that the population doesn't distribute normal within themselves (subsection “NGF induces production of PIP3 by PI3K followed by trafficking of TRPV1 channels to the PM”), however they used a parametric test for paired data that assumes normal distribution. Moreover, how the authors deal with outliers? Potential outliers are visible in Figure 2D, Figure 3C and Supplementary Figure 5B.

We thank the reviewer for pointing this out. We have redone the analysis using non-parametric statistical tests, described in the Materials and methods. All outliers were included in the analysis.

Reviewer #3:[…] 1) The authors should determine the TRPV1-PI3K interaction site (currently located within ~248 residues of the ARD), since they have a good readout with the TRPV1-ARD experiments (as shown in Figure 5B). I suggest performing a sequence alignment and analyze the conserved residues between channels; this analysis can be used to generate new TRPV1-ARD constructs that might lack the potentiation effect and determine the TRPV1-PI3K interaction site. This is important since targeting this site could help modulate the NGF-TRPV1 mediated sensitization.

See response to Editor point #1.

2) The authors should repeat the experiment shown in Figure 6B using a mammalian TRPA1 instead of the zebrafish one, as these channels would likely display higher expression levels in F11 cells. This would help determining whether TRPA1 significantly potentiates NGF‐induced PI3K activity.

As part of the refocusing of the manuscript on TRPV channels, the TRPA1 data in question were removed. See response to Editor point #1.

3) In Figure 5—figure supplement 2A, the authors should increase the number of samples for TRPV4 (equivalent to the other channels). In its present form, most of the data points lie within the distribution of the control.

We performed statistical tests to determine whether the dataset for TRPV4 supported the assertion that TRPV4 enhanced NGF-induced PI3K activity compared to control cells and found there to be a statistically significant difference.

4) The authors should include in Figure 5C the data points corresponding to the control and TRPV1.

We have modified the figure as requested.

5) Although the experiments with TRPM4 and TRPM8 show that they do not have reciprocal regulation, both channels display opposite effects for PI3K activity and channel trafficking (Figures 6B and 6D). The authors should provide a deeper discussion for this opposite effect.

As part of the refocusing of the manuscript on TRPV channels, the TRPM data in question were removed. See response to Editor points #1 and #3.

6) Testing channels (TRPM) without ARD is a good idea; however, coming back to suggestion #1, I think that finding a TRPV1 and/or TRPV1-ARD construct that decouples potentiation of PI3K activity from the expression will support the model proposed in Figure 7.

See response to Editor point #1.

[Editors’ note: the author responses to the re-review follow.]

Essential revisions:1) The Akt-PH domain is a combined PI(3,4,5)P3 and PI(3,4)P2 sensor. The authors argued that their Akt phosphorylation assay is an independent readout of PI(3,4,5)P3 production; therefore there is no need to repeat the experiment with a more specific PI(3,4,5)P3 sensor. I disagree with this argument. My understanding of Akt regulation is that this enzyme binds to the plasma membrane via its PH domain that binds both PI(3,4,5)P3 and PI(3,4)P2, and as a result it becomes phosphorylated by PDK1, see Balla Physiological Reviews 2013. See also Franke et al. Science 1997 PMID:9005852, showing that PI(3,4)P2 stimulates Akt. In other words, Akt phosphorylation is also a combined PI(3,4,5)P3 PI(3,4)P2 assay.As the data stands, everything can be explained by a potentiated increase in PI(3,4)P2, which is also increased upon PI3K stimulation. This is not a very likely scenario of course, but it is very easy to test. I strongly recommend that the authors perform the experiment with a more specific PI(3,4,5)P3 sensor.

Our work revealing TRPV1 potentiation of NGF-induced PI3K activity used Akt-PH as a phosphoinositide probe. The reviewer correctly points out that Akt-PH has similar apparent affinities for PI(3,4)P_2_ and PI(3,4,5)P_3_. Indeed, both phosphoinositides are known products of PI3K and both are known to mediate downstream signaling through phosphorylation of the protein kinase Akt. The Reviewer asked that we use the more specific PI(3,4,5)P_3_ probe Btk-PH to determine whether TRPV1-potentiation of NGF-induced PI3K activity produces PI(3,4)P_2_ or PI(3,4,5)P_3_.

We repeated our NGF signaling experiments using Btk-PH (fused to CFP) as a phosphoinositide probe. We found that expression of Btk-PH in our cells completely prevented the NGF-induced increase of TRPV1 to the plasma membrane (the data are now included in Figure 1—figure supplement 1). This is not surprising, as Btk-PH has been reported to interfere with PI3K signaling due to sequestration of PI(3,4,5)P_3_ (Varnai et al., 2005). Because Btk-PH blocked NGF signaling to TRPV1 in our system, we could not use it to test whether the increased phosphoinositide production in the presence in TRPV1-expressing cells compared to control cells was due to PI(3,4)P_2_ or PI(3,4,5)P_3_. The conclusion of our paper, that TRPV1 potentiates NGFinduced PI3K activity, is not affected by this result, although of course we are disappointed to be unable to fully satisfy the reviewer.

Although Btk-PH blocked NGF signaling to TRPV1, we did test whether it also blocked TRPV1 potentiation of NGF-induced PI3K activity. As shown Figure 1—figure supplement 1, Btk-PH indeed blocked this effect as well. Thus, Btk-PH blocked both parts of the reciprocal regulation among PI3K and TRPV1: it prevented PI3K signaling to increase trafficking of TRPV1 to the plasma membrane and it prevented TRPV1 signaling to enhance the activity of PI3K.

We have made several modifications to the manuscript to reflect the reviewer’s suggestions, including: the new data on Btk-PH are shown in Figure 1—figure supplement 1; Akt-PH is referred to as a PI(3,4)P_2_/PI(3,4,5)P_3_ probe; and citations of PI(3,4)P_2_ as a product of PI3K and a mediator of Akt phosphorylation are now included in the text.

2) My original request was to test the effect of wortmannin on the translocation of the Akt-PH domain, not that of TRPV1. Again, this is an extremely simple experiment, and would solidify the key finding.

We performed the requested experiment examining the effect of wortmannin on NGF signaling. We found that wortmannin (20 nM) prevented the NGF-induced translocation of Akt-PH to the plasma membrane and increased trafficking of TRPV1 in both control cells (Figure 2—figure supplement 2) and TRPV1-expressing cells (Figure 1C and D). These data are now discussed in the main text.

3) The authors forgot to delete the second half of the following sentence from the Abstract: "Further, other TRPV channels with conserved ARDs also potentiated NGF‐induced PI3K activity whereas TRP channels lacking ARDs did not." Also the effects of TRPV4 and TRPV2 on Akt-PH translocation look smaller than that induced by TRPV1, are the effects significantly different from TRPV1? I recommend that this is tested, and shown in Table 1 and mentioned in the Results.

We have made the requested revision in the Abstract. We now include statistical significance of Akt-PH translocation amplitudes with TRPV2 and TRPV4 compared to TRPV1 in Table 1. We now discuss these comparisons in the main text.